# Amide C–N bonds activation by A new variant of bifunctional N-heterocyclic carbene

Yuxing Cai[1,2,4], Yuxin Zhao[3,4], Kai Tang[2], Hong Zhang[2], Xueling Mo[2], Jiean Chen ®[2] ✉ & Yong Huang ®[3] ✉

We report an organocatalyst that combines a triazolium N-heterocyclic carbene (NHC) with a squaramide as a hydrogen-bonding donor (HBD), which can effectively catalyze the atroposelective ring-opening of biaryl lactams via a unique amide C–N bond cleavage mode. The free carbene species attacks the amide carbonyl, forming an axially chiral acyl-azolium intermediate. Various axially chiral biaryl amines can be accessed by this methodology with up to 99% ee and 99% yield. By using mercaptan as a catalyst turnover agent, the resulting thioester synthon can be transformed into several interesting atropisomers. Both control experiments and theoretical calculations reveal the crucial role of the hybrid NHC-HBD skeleton, which activates the amide via H-bonding and brings it spatially close to the carbene centre. This discovery illustrates the potential of the NHC-HBD chimera and demonstrates a complementary strategy for amide bond activation and manipulation.

The amide bond is a ubiquitous and vital structural motif in biological and chemical systems, as it constitutes the backbone of peptides, proteins, pharmaceuticals and polymers[1–5]. Developing efficient and selective methods for the formation and cleavage of amide species is a primary research goal in various fields of chemistry and biology[6–9]. However, the amide C–N bond cleavage is exceptionally challenging owing to the resonance stabilization (19-26 kcal mol$^{-1}$) of the amide group[10–12]. Most current catalytic methods for this transformation employ transition metal complexes as catalysts[7–9,13–15], whereas only a few examples use organocatalysts that activate amides through a dual hydrogen-bonding mode[16,17]. Nevertheless, hardly any method can achieve a direct nucleophilic substitution at the amide carbon to break the C–N bond. The main reason for this is the relatively short C–N bond distance (about 1.3 Å), which prevents the access of organocatalysts to the reaction centre[11]. N-heterocyclic carbene (NHC) is a promising organocatalyst owing to its high nucleophilicity and ability to activate carbonyl compounds[18–21]. Various acyl derivatives, such as anhydrides, imides and esters, have been successfully transformed by NHC-mediated catalysis, forming acyl-azolium intermediates via

nucleophilic substitution mechanisms[22–27]. However, the application of NHCs to amide substrates has not been reported yet (Fig. 1a). In 2005, Movassaghi et al.[28] disclosed an amidation protocol for unactivated esters using hydroxy-tethered amines as transacylation agents. This method involves a non-covalent hydroxy activation for an initial transesterification followed by a fast O-to-N acyl transfer. This example implies that the amide C–N bond is resistant to cleavage even in the presence of intramolecular hydroxy groups and free carbene species.

To address this problem, we explore two aspects. On the one hand, we combine the axial chiral construction logic and use lactam as the amide precursor, introducing ring strain to lower the activation energy barrier of the amide bond itself[16,17,29–31]. In recent years, NHC-catalyzed axially chiral molecules have made significant progress[32,33]. On this basis, we aim to broaden this area to an atroposelective ring-opening scenario (Fig. 1b). On the other hand, the chimera strategy of fusing NHC with an H-bonding donor (HBD) has emerged as an intriguing and influential research direction[34–36]. The simultaneous non-covalent interaction of HBD with the substrate effectively lowers the activation barrier and stabilizes a more spatially-ordered

[1]State Key Laboratory of Chemical Oncogenomics, Peking University Shenzhen Graduate School, 518055 Shenzhen, China. [2]Pingshan Translational Medicine Center, Shenzhen Bay Laboratory, 518118 Shenzhen, China. [3]Department of Chemistry, The Hong Kong University of Science and Technology, Clear Water Bay, Kowloon, Hong Kong SAR, China. [4]These authors contributed equally: Yuxing Cai, Yuxin Zhao. ✉e-mail: chenja@szbl.ac.cn; yonghuang@ust.hk

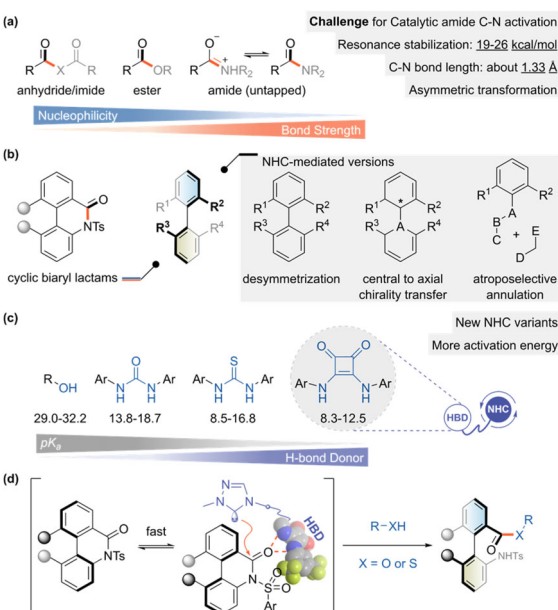

**Fig. 1 | Catalytic amide C–N bond cleavage via bifunctional NHCs. a** NHC activation for the non-aldehyde substrate. **b** Atroposelective ring-opening activation of amide. **c** Integration with non-covalent activation module. **d** NHC-mediated C–N activation (this work).

transition state[37–42]. Connon's group designed an amide-tethered triazolium NHC for an asymmetric benzoin condensation reaction[43]. Ye's group introduced a new series of NHC derivatives bearing a free hydroxyl group and successfully achieved various enantioselective transformations[35,44,45]. Recently, our group showed that by incorporating a tethered urea or thiourea moiety, the NHC catalysts could enable a new range of asymmetric reactions[46,47]. To enhance the H-bonding donor ability, we hypothesized that squaramide could be attached to the NHC framework to activate the amide for C–N bond cleavage (Fig. 1c)[37,40,42]. Based on this hypothesis, we designed and synthesized an aminoindanol-derived triazolium NHC catalyst fused with a squaramide unit and applied it in dynamic kinetic resolution (DKR) of cyclic biaryl lactams[16,48]. The preliminary mechanistic studies by theoretical calculations and control experiments confirmed the essential role of H-bonding interaction between the amide substrate and the HBD moiety, and also verified the direct cleavage of the amide C–N bond by nucleophilic substitution of the free NHC species. This reaction exhibits mild conditions and broad substrate scope, affording atropisomeric biaryls. It also achieves the N-to-O/N-to-S acyl transfers as less conventional conversion modes (Fig. 1d).

## Results and discussion

We initiated our investigation with cyclic biaryl lactam **1a** and benzyl alcohol **2a** in the presence of the NHC precursor **3** (Table. 1). The reaction using $C_2$-symmetric imidazolium pre-catalyst **3a** gave a low enantioselectivity of 2% ee, although an 85% yield was obtained. The aminoindanol-derived triazolium NHC pre-catalyst **3b** afforded the desired product in only 56% yield and 2% ee. In contrast, the Waser-type bifunctional NHC pre-catalyst **3c** achieved moderate enantioselectivity with good conversion (70%, −36% ee)[34,49]. The enantiocontrol was significantly improved when a squaramide unit was first fused with NHC on the same skeleton (cat. **3d**, 99%, −66% ee). This suggested that the chimeric NHCs might be effective for the C–N cleavage of lactam and that the enhanced H-bonding strength might improve both the reactivity and enantiocontrol. We then evaluated a series of aminoindanol-derived NHC-thiourea chimeras and found that the 3,5-(CF$_3$)$_2$-phenyl substituted pre-catalyst **3e** gave an excellent result

(99%, 94% ee). We also tried to further increase the H-bonding donor ability by replacing the thiourea with a selenourea analogue **3j**, which could roughly maintain the reaction performance (99%, 92% ee). When the squaramide group was used as an HBD moiety (**3k**), the enantiomeric excess slightly increased to 95%, and it was chosen as the optimal catalyst. A solvent, base and temperature screening indicated that the standard conditions in entry 7 gave the best outcome.

### Substrate scope

We then explored the substrate scope of alcohols under the optimized reaction conditions. As shown in Fig. 2, various substituted benzyl and heteroaryl alcohol derivatives efficiently afforded the corresponding products with high yields and excellent enantioselectivities (**4a-4f**). Primary alcohols, including the simplest methanol and ethanol, were suitable catalyst turnover agents (**4g-4k**). However, secondary and tertiary alcohols failed to complete the catalytic cycle mainly owing to steric hindrance. Some functional groups, such as cyclopropane, primary halides, and ester, were well-tolerated under the reaction conditions (**4k-4n**). Unsaturated bonds attached to alcohols did not affect the reaction performance (**4q-4s**). When a diol was used, exclusive chemoselectivity of the primary hydroxyl end was observed, which could be rationalized by the kinetic factor (**4t**). The allyl alcohol with a long flexible chain, phytol, was also a compatible substrate for the target atropisomeric biaryl with excellent yield and ee value (**4v**). The crystal structure of compound **4h** was determined by X-ray diffraction, confirming the axially chiral biaryl with *R*-configuration.

We then examined the scope of cyclic biaryl lactams (Fig. 3). Cyclic biphenyl lactams bearing different substituents (Me, OMe or OBn) on either aromatic ring gave the target products nearly quantitative yields and ee values ranging from 86% to 99% (**5a-5i**). The rotational barrier did not require substituents on both 2'- and 6-positions, as a 2',5-disubstituted analogue could still afford the desired product and maintain conformational stability (**5 g**, 98%, 96% ee). Two more complex moieties were also attached to the lactam skeletons, showing a slight decrease in the reaction performance, which further demonstrated the functional group tolerance (**5j** and **5k**). This atroposelective ring-opening protocol could also handle the phenyl-naphthyl- or phenyl-benzofuranyl-type lactams, leading to similar outcomes for both reaction efficiency and enantiocontrol (**5l-5o**).

Mercaptans could also act as catalyst turnover agents to afford various thioester products (Fig. 4). Benzyl thiols bearing electron-donating groups on the aromatic ring were well tolerated to give both high yield and enantioselectivities; in contrast, the electron-withdrawing group tethered analogue declined on both counts mainly owing to the occurrence of side reaction (**6a-6d**). Both primary and secondary thiols were compatible with the presented protocol, affording the corresponding axially chiral biaryl compounds with satisfactory results (**6e-6h**). When cholesterol was used, slightly lower enantioselectivity was obtained due to the chirality mismatch of the secondary thiol (**6i**, 95%, 86% de). The structural variations of the biaryl scaffold were also examined, most of which efficiently gave the desired products in good yields with enantiomeric excesses ranging from 88% to 99% (**6j-6s**). Attempts to construct atropisomers with complex skeletons showed that the reaction efficiency and enantioselectivity were slightly reduced; however, it still demonstrated the potential as a late-stage modification method (**6t-6w**).

### Synthetic applications and mechanistic studies

We then scaled up the model reaction tenfold to demonstrate the synthetic utility of the protocol, which still performed well (product **6r**, 74%, 92% ee). The resulting thioester synthon could be reduced by Et$_3$SiH/Pd or NaBH$_4$, affording axially chiral aldehyde **7a** or alcohol **7b** with preserved chirality. It also enabled rapid access to the

## Table. 1 | Optimization of the reaction conditions

| Entry | Catalyst | Solvent | Base | T (°C) | yield (%) | ee (%) |
|---|---|---|---|---|---|---|
| 1 | **3k** | *n*-hexane | LiHMDS | r.t. | 99 | 66 |
| 2 | **3k** | toluene | LiHMDS | r.t. | 94 | 62 |
| 3 | **3k** | DMSO | LiHMDS | r.t. | 94 | 76 |
| 4 | **3k** | MTBE | LiHMDS | r.t. | 99 | 75 |
| 5 | **3k** | CDCl$_3$ | LiHMDS | r.t. | 99 | 76 |
| 6 | **3k** | DCM | LiHMDS | r.t. | 88 | 78 |
| 7 | **3k** | DCM | LiHMDS | −20 | 99 | 95 |
| 8 | **3k** | DCM | DBU | −20 | 93 | 94 |
| 9 | **3k** | DCM | DIPEA | −20 | 99 | 92 |
| 10 | **3k** | DCM | Et$_3$N | −20 | 99 | 90 |
| 11 | **3k** | DCM | K$_2$CO$_3$ | −20 | 89 | 84 |
| 12 | **3k** | DCM | K$_3$PO$_4$ | −20 | 99 | 74 |

Reaction conditions: cyclic biaryl lactam **1a** (0.05 mmol), benzyl alcohol **2a** (0.1 mmol), NHC precursor **3** (20 mol%), and base (16 mol%) were stirred in a solvent (0.5 mL) at −20 °C under argon for 15 h. Yields were determined by NMR using 1,3,5-trimethoxybenzene as an internal standard. The enantiomeric excess (ee) was determined by chiral HPLC.

oxadiazole motif **7c** via a one-pot protocol (Fig. 5a). We then designed some experiments for mechanistic elucidation based on the concept of an acyl-azolium intermediate. The high-resolution mass spectroscopy (HRMS) analysis showed that the exact mass for the direct adduct of lactam **1a** and bifunctional NHC catalyst **3k** could be detected (Fig. 5b, left). This suggested that the free carbene could initiate the amide C−N bond cleavage via nucleophilic attack, forming the chirality-determining acyl-azolium intermediate. Meanwhile, the mass spectrum signal for the protonation of nitrogen anion was observed, indicating the covalent linkage mode between the lactam substrate and NHC catalyst (Fig. 5b, right). Our previous reports have shown that a proton shuttle might operate when the protonation step is a critical catalytic step. Based on this hypothesis, we tried introducing H$_2$O or PhCO$_2$H as an additive and found that both could improve the enantioselectivity (Fig. 5c). Even 10.0 equivalents of H$_2$O could be tolerated to give a similar result. In contrast, the acidic additive did not show such high compatibility. Moreover, the conformational effect of the amide on the C−N bond cleavage reaction was investigated. The control experiment showed that the torsional strain induced by the 2′,6-disubstituted pattern was an essential activation factor. Under standard conditions, the reaction was completely suppressed for unsubstituted cyclic biaryl lactams. The C−N

bond also remained intact during the reaction period for acyclic amide (Fig. 5d).

The hybrid skeleton composed of triazolium NHC and squaramide as HBD was examined. As shown in Table. 2, the simultaneous use of conventional triazolium NHC **3b** with squaramide as a separate HBD additive could significantly increase the conversion from 56% to 98%, compared with the case without HBD additive (entries 2 and 3). Adding bifunctional NHC catalyst **3k** or squaramide alone did not trigger the reaction (entries 4–6). A catalytic amount of LiHMDS could only afford a similar ring-opening product **4a**, failing to achieve the catalytic version of the whole reaction cycle (entry 7). The result confirmed the unique activation ability of the newly designed NHC-HBD chimera, offering completely different catalytic properties from either NHC or HBD individual components.

Based on the above mechanistic studies, we proposed the catalytic cycle of the NHC-HBD chimera catalyzed C−N bond cleavage of lactam (Fig. 6). The deprotonated carbene pre-catalyst initially interacted with lactam via H-bonding interaction. Theoretical calculations were then performed to determine which mode was more favorable. Three possible forms of interaction were considered, and the diplex H-bonding with amide carbonyl was slightly preferred (**Int-1A**, −5.4 kcal mol$^{-1}$; **Int-1B**, −5.3 kcal mol$^{-1}$; **Int-1C**, −4.5 kcal mol$^{-1}$).

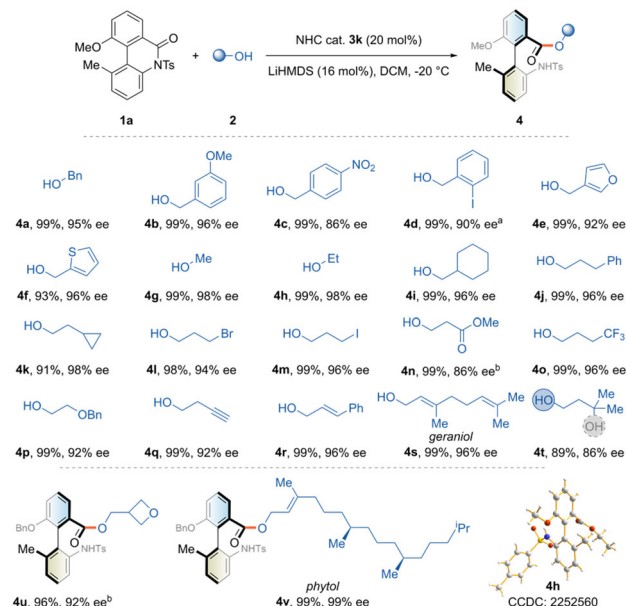

**Fig. 2 | The reaction scope of alcohols.** Reactions were performed using cyclic biaryl lactams **1** (0.05 mmol), alcohols **2** (0.1 mmol), NHC precursor **3k** (20 mol%), and LiHMDS (16 mol%) in DCM (0.5 mL) at −20 °C under argon for 15 h. Yields refer to isolated products. Ee was determined by chiral HPLC. ᵃCyclic biaryl lactam **1b** was used as substrate. ᵇAfter stirring for 15 hours at −20 °C, the reaction period was prolonged for another 5 h at 0 °C. See Supplementary Methods for experimental details.

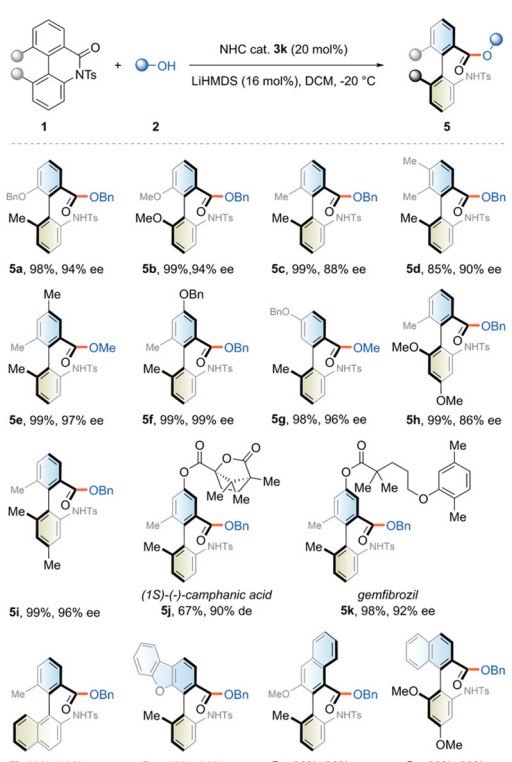

**Fig. 3 | The scope of cyclic biaryl lactams.** See Supplementary Methods for experimental details.

The energy barriers for transition states guided by these intermediates have a more pronounced difference, further verifying **Int-1A** as the reaction intermediate with an energy advantage (**TS-1A**, 19.3 kcal mol⁻¹; **TS-1B**, 24.2 kcal mol⁻¹; **TS-1C**, 37.3 kcal mol⁻¹). It resulted in spatial

**Fig. 4 | The scope of thiols and cyclic biaryl lactams.** See Supplementary Methods for experimental details.

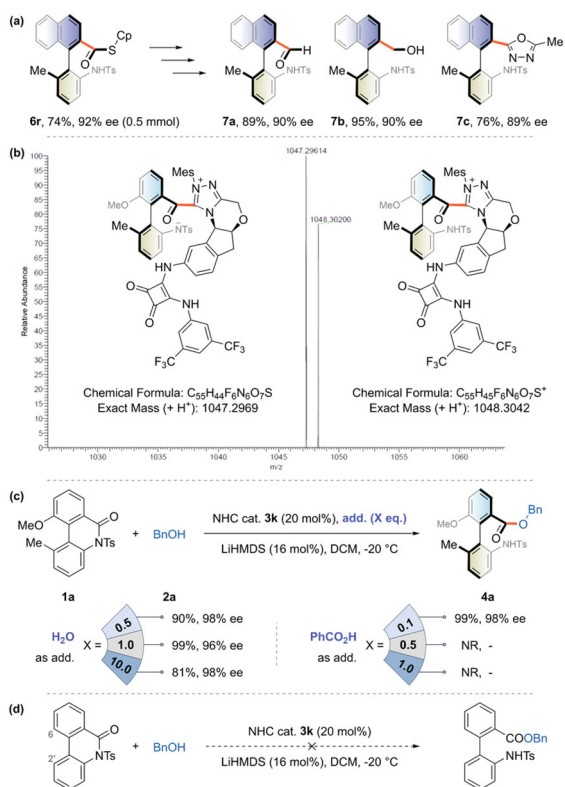

**Fig. 5 | Synthetic application and mechanism study. a** Synthetic applications of axially chiral biaryl thioester. **b** HRMS analysis of intermediate species. **c** proton shuttle experiments. **d** Control experiments.

## Table. 2 | The control experiments

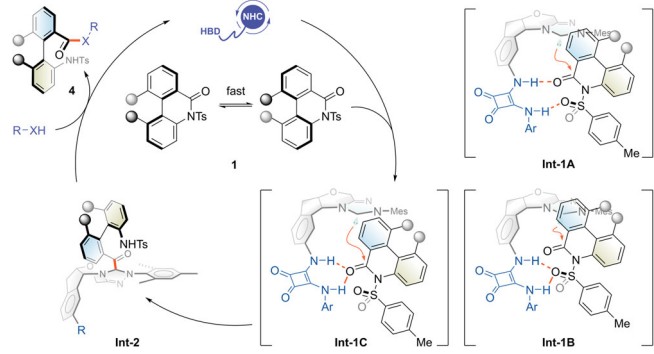

| Entry | Catalyst | Base | Additive | Yield (%) | ee (%) |
|---|---|---|---|---|---|
| 1 | 3k | LiHMDS | – | 99 | 95 |
| 2 | 3b | LiHMDS | – | 56 | 2 |
| 3 | 3b | LiHMDS | √ | 98 | 4 |
| 4 | – | – | √ | NR | – |
| 6 | 3k | – | – | NR | – |
| 5 | – | – | – | NR | – |
| 7 | – | LiHMDS | - | 56 | – |

Reaction conditions: cyclic biaryl lactam **1a** (0.05 mmol), benzyl alcohol **2a** (0.1 mmol), NHC precursor **3** (20 mol%), LiHMDS (16 mol%), and additive (20 mol%) were stirred in DCM (0.5 mL) at −20 °C under argon for 15 h. Yields were determined by NMR using 1,3,5-trimethoxybenzene as an internal standard. The enantiomeric excess (ee) was determined by chiral HPLC.

**Fig. 6 | Proposed catalytic cycle for the ring-opening reaction.** See Supplementary Methods for experimental details.

alignment for the nucleophilic carbene centre to attack the amide, followed by a protonation step to afford the axially chiral acyl-azolium intermediate **Int-2**. Finally, acyl transfer by nucleophilic reagents regenerated the NHC catalyst and gave the ring-opening product **4**. In this stage, other possible non-covalent interactions between catalyst and substrate skeletons are not involved or discussed[50].

In summary, we have developed a bifunctional chimera combining triazolium NHC with squaramide as HBD, which was shown to be effective in the atroposelective ring-opening of biaryl lactams. This organocatalytic protocol formally achieved a unique amide C−N bond cleavage mode via nucleophilic attack of free carbene species. Various axially chiral biaryl amines could be readily accessed by the proposed methodology with up to 99% ee and 99% yield. By using mercaptan as a catalyst turnover agent, the resulting thioester synthon could be generated and quickly transformed into several interesting atropisomers. Both control experiments and theoretical calculations revealed the crucial role of the hybrid NHC-HBD skeleton. The squaramide moiety initially activated the amide via H-bonding, bringing it spatially close to the carbene centre. The targeted C−N bond broke via a direct NHC nucleophilic attack on the amide carbonyl. The present discovery illustrates the potential of

the NHC-HBD chimera, and further application scenarios are under investigation in our laboratory.

## Methods

### General method for the NHC-HBD catalyzed atroposelective ring-opening of biaryl lactams

The catalyst precursor **3k** (7.4 mg, 0.01 mmol, 20 mol%) and cyclic biaryl lactam **1a** (19.7 mg, 0.05 mmol, 1.0 eq.) were mixed in anhydrous DCM (0.5 mL, 0.1 M) in an oven-dried test tube (20-mL). The mixture was degassed and backfilled with argon (3x) before adding LiHMDS (1.0 M in THF, 8 μL, 16 mol%). The test tube was stirred at −20 °C for 10 min. Benzyl alcohol **2a** (10 μL, 0.1 mmol, 2.0 eq.) was directly added, and the mixture was stirred at -20 °C for 15 hours. Upon complete consumption of **1a**, the reaction was purified by flash column chromatography (eluent: PE/EA = 10/1 to 4/1) to afford product **4a** as a white solid (24.8 mg, 99% yield, 95% ee). The ee was determined by chiral HPLC, conditions: Chiralpak-AS-H column, hexane/iPrOH = 95/5, 1.0 mL/min: $t_{major}$ = 21.300 min; $t_{minor}$ = 24.717 min. [α] = +5.0 (c = 0.5 in $CHCl_3$). **m.p.** 81-82 °C. **$^1$H NMR** (400 MHz, $CDCl_3$) δ 7.59 (dd, $J$ = 7.9, 5.6 Hz, 3H), 7.46 (t, $J$ = 8.1 Hz, 1H), 7.38 (d, $J$ = 8.2 Hz, 1H), 7.32−7.22 (m, 3H), 7.15 (d, $J$ = 8.0 Hz, 2H), 7.13 − 7.00 (m, 4H), 6.83 (d, $J$ = 7.6 Hz, 1H), 6.18 (s, 1H), 4.97 (d, $J$ = 12.2 Hz, 1H), 4.80 (d, $J$ = 12.3 Hz, 1H), 3.54 (s, 3H), 2.34 (s, 3H), 1.72 (s, 3H). **$^{13}$C NMR** (101 MHz, $CDCl_3$) δ 166.5, 157.1, 143.4, 137.5, 137.3, 135.3, 134.7, 132.8, 130.1, 129.5, 128.5, 128.4, 128.3, 128.1, 127.6, 127.5, 125.8, 124.9, 123.0, 116.8, 114.9, 67.1, 56.1, 21.6, 20.2. **HRMS** (ESI-TOF) [M + H]$^+$ calculated for [$C_{29}H_{28}NO_5S$]$^+$ 502.1683, found 502.1684.

## Data availability

The X-ray structure data generated in this methodology have been deposited in the Cambridge Crystallographic Data Centre (CCDC: 2252560 for **4h**, 2263709 for **7b**). Copies of the data can be obtained free of charge via https://www.ccdc.cam.ac.uk/structures/. Experimental procedures, characterizations of new compounds and DFT calculation results are included in the Supplementary Methods. For NMR and HPLC spectra of structurally novel compounds, see Supplementary Figures. All other data are available from the authors upon request.

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

## Acknowledgements

This work was financially supported by the National Natural Science Foundation of China (21825101, Y.H.), Hong Kong RGC (16300320, Y.H.), Shenzhen Science and Technology Innovation Commission (SGDX2019081623241924, Y.H.; KCXFZ20201221173404013, J.C.). We are grateful to the Shenzhen Bay Laboratory Supercomputing Center for the assistance in DFT calculation.

## Author contributions

Y.H. and J.C. conceived and directed the project. Y.C., Y.Z., K.T., H.Z. and X.M. performed the experiments and analyzed the experimental data. K.T. performed the DFT calculations. Y.H. and J.C. wrote the manuscript with input from all authors. All authors have read and approved the final manuscript.

## Competing interests

The authors declare no competing interests.
