## [Peer Review File · Nature Communications]

Amide C-N Bonds Activation by A New Variant of Bifunctional N-Heterocyclic CarbeneREVIEWER COMMENTS

Reviewer #1 (Remarks to the Author):

Chen, Huang and co-workers report "Amide C-N Bonds Activation by A New Variant of Bifunctional N-Heterocyclic Carbene". Strained biphenyl-based lactams that readily undergo racemization are enantioselectively ring-opened with various primary alcohols and thiols to give the corresponding esters and thioesters in excellent yields and very good to excellent enantioselectivities (dynamic kinetic resolution). Authors developed an aminoindanol-derived bifunctional NHC catalyst that carries a squaramide H-bonding activator moiety. Other bifunctional catalysts tested delivered worse results and the necessity of the bifunctional system was addressed by subjecting the individual catalyst entities to the test reaction. In both cases without any success, clearly indicating that the covalent nature of the bifunctional system is of importance. The scope of the reaction is well documented and mechanistic studies are provided. The asymmetric synthesis of axially chiral compounds is currently heavily investigated. I therefore see the paper located in a hot field. Results are very good in my eyes and the paper is well written. Although NHC-catalysis has been applied to the synthesis of axially chiral compounds through the use of chiral acylazolium ions as acylation reagents, the herein applied approach is conceptually different. Therefore, I support publication in Nature Commun. subject to the following modifications:

A) In the introduction the authors should better discuss the current strategies/methods known in the area of NHC catalyzed atroposelective synthesis of axially chiral compounds (include in Figure 1).

B) The scope is rather well addressed; however, as the leaving group always the N-Tos moiety is used. Any reason? Is the Tos-group of importance; e.g. does NMe₂ also work? I also assume that the corresponding lactones do not work, likely too reactive and back ground reaction is a problem. Please comment.

C) Further, I am wondering whether primary amines (maybe beta-branched to reduce nucleophilicity) can be used as nucleophiles, or is the non-catalyzed lactam-opening in such cases too fast?

D) The addition of the chiral NHC to the lactam is the stereodetermining step. I ask authors to design a lactam with bulkier ortho,ortho'-substituents where racemization under the reaction conditions is slow or ideally fully suppressed. This lactam should then engage in a

kinetic resolution with the designed catalyst, which would further expand the applicability of the method.

Reviewer #2 (Remarks to the Author):

The authors have developed an aminoindanol-derived triazolium NHC catalyst functionalized with a squaramide unit and applied it in the dynamic kinetic resolution of cyclic biaryl lactams. While the scope is somewhat limited to specific amides, the strategy is intriguing, and the methodology proves effective. I believe this work will appeal to those interested in organocatalysis, noncovalent interactions, and the synthesis of atropisomeric compounds.

Minor Revision:

Regarding non-covalent interactions (NCIs): NHCs have been extensively studied for their effective non-covalent interactions with carbonyl compounds, as discussed in ACS Catalysis 2023, 13, 407. This includes H-bonds, face-to-edge interactions, stacking, and others, which are closely related to the central concepts of this work. I recommend the authors cite this work for a more comprehensive discussion in the introduction.

Reviewer(s)' Comments to Author (Responses are highlighted in red):

Reviewers' Comments:

Reviewer #1: Chen, Huang and co-workers report "Amide C-N Bonds Activation by A New Variant of Bifunctional N-Heterocyclic Carbene". Strained biphenyl-based lactams that readily undergo racemization are enantioselectively ring-opened with various primary alcohols and thiols to give the corresponding esters and thioesters in excellent yields and very good to excellent enantioselectivities (dynamic kinetic resolution). Authors developed an aminoindanol-derived bifunctional NHC catalyst that carries a squaramide H-bonding activator moiety. Other bifunctional catalysts tested delivered worse results and the necessity of the bifunctional system was addressed by subjecting the individual catalyst entities to the test reaction. In both cases without any success, clearly indicating that the covalent nature of the bifunctional system is of importance. The scope of the reaction is well documented and mechanistic studies are provided. The asymmetric synthesis of axially chiral compounds is currently heavily investigated. I therefore see the paper located in a hot field. Results are very good in my eyes and the paper is well written. Although NHC-catalysis has been applied to the synthesis of axially chiral compounds through the use of chiral acylazolium ions as acylation reagents, the herein applied approach is conceptually different. Therefore, I support publication in Nature Commun.

Our response: We would like to thank reviewer 1 for the positive comments on the overall novelty and quality of the manuscript. Recently, we have focused on the bi-functional NHC catalyst design and attempted to expand the possible scenarios. This report demonstrated its efficacy in atroposelective ring-opening activation of amide, and on this basis, we believe that bifunctional catalysts are likely to achieve other chiral transformations that have not yet been reported.

Technical questions:

1. In the introduction the authors should better discuss the current strategies/methods known in the area of NHC catalyzed atroposelective synthesis of axially chiral compounds (include in Figure 1).

Our response: We appreciate reviewer 1 for his/her professional suggestion. In the previous version, the introduction lacked a transitional discussion about the methodology design. This part has been revised, and a brief discussion of the current strategies in NHC-mediated atroposelective synthesis of axially chiral compounds was appended in **Scheme 1 B**. Also, some representative references were supplemented (Ref. 29-33).

2. The scope is rather well addressed; however, as the leaving group always the N-Tos moiety is used. Any reason? Is the Tos-group of importance; e.g. does NMe₂ also work? I also assume that the corresponding lactones do not work, likely too reactive and back ground reaction is a problem. Please comment.

Our response: We thank reviewer 1 for his/her questions. As judged by the reviewer, the N-Tos group is indeed indispensable for both reactivity and enantioselectivity. As for NMe₂, although the conversion was good, the enantiocontrol decreased dramatically (100% conversion, 50% ee). Other amine-protecting groups, such as Boc, Ns and Mes, are incompatible with the standard conditions. As discussed in the manuscript, the NHC-HBD catalyst would form a diplex H-bonding with amide carbonyl, which is essential for initiating the ring-opening process. The other substituent on N-atom

can directly affect the strength of this interaction. In addition, there is another critical equilibrium regarding the leaving ability of N-PG and the steric hindrance it can cause, determining whether this reaction can be efficiently carried out with facial discrimination. Overall, the N-Tos moiety has become a suitable choice at this stage.

In addition, as the reviewer suspected, we carried out a series of related experiments about cyclic biaryl lactones with different nucleophiles in the presence of NHC-HBD catalysts. Taking BnNH_2 as an example, the background reaction causes much trouble until the temperature decreases to $-60\text{ }^\circ\text{C}$. Under this condition, we screened a lot of reaction parameters, such as catalysts, bases, additives and solvents. Up to now, we have obtained 90% yield and 64% ee value as optimal results. With benzyl alcohol as a nucleophilic partner, the reaction can also obtain a 69% yield and 74% ee. In this stage, benzyl mercaptan did not exhibit a similar reactivity match. At present, our laboratory is studying the corresponding solution.

3. Further, I am wondering whether primary amines (maybe beta-branched to reduce nucleophilicity) can be used as nucleophiles, or is the non-catalyzed lactam-opening in such cases too fast?

Our response: We thank reviewer 1 for his/her questions. What the reviewer mentioned was another critical point of this reaction. As discussed above, the reactivity of the turnover reagent should also achieve a delicate balance to be struck with the various steps of the catalytic cycle. This phenomenon is consistent with previous reports on NHC-mediated systems. We investigated primary amines involving alpha-branched types, such as benzhydrylamine and cycloheptylamine, and the results were unsatisfactory. Even with diphenylamine, the reaction is still uncontrollable for enantiocontrol. We hypothesized that the background reaction was too rapid to realize the facial discrimination, as the

reviewer judged.

4. The addition of the chiral NHC to the lactam is the stereodetermining step. I ask authors to design a lactam with bulkier ortho,ortho'-substituents where racemization under the reaction conditions is slow or ideally fully suppressed. This lactam should then engage in a kinetic resolution with the designed catalyst, which would further expand the applicability of the method.

Our response: We appreciate reviewer 1 for his/her professional suggestion. According to the previous research on cyclic biaryl lactones (Ref: *Angew. Chem. Int. Ed.* **2005**, *44*, 5384), the rotational barriers of the lactones were investigated, and only bulky groups can suppress the atropisomerization. We designed a bulky cyclic biaryl lactam, which was speculated to have a long enough half-life (R¹, R³ = ^tBu). Unfortunately, we face technical difficulties in preparing the target compound. During the synthetic condition screening, we found that the substituents with considerable steric hindrance at the ortho-position dramatically affect the stability of the lactam skeleton—most of the reactions became messy during the reaction period. We are still considering implementing this proposal, and the atroposelective ring opening of the reported lactone under our conditions may be a preliminary template reaction.

Reviewer #2: The authors have developed an aminoindanol-derived triazolium NHC catalyst functionalized with a squaramide unit and applied it in the dynamic kinetic resolution of cyclic biaryl lactams. While the scope is somewhat limited to specific amides, the strategy is intriguing, and the methodology proves effective. I believe this work will appeal to those interested in organocatalysis,

noncovalent interactions, and the synthesis of atropisomeric compounds.

Our response: We thank reviewer 2 for his/her positive comments on the overall report.

1. Regarding non-covalent interactions (NCIs): NHCs have been extensively studied for their effective non-covalent interactions with carbonyl compounds, as discussed in ACS Catalysis 2023, 13, 407. This includes H-bonds, face-to-edge interactions, stacking, and others, which are closely related to the central concepts of this work. I recommend the authors cite this work for a more comprehensive discussion in the introduction.

Our response: We appreciate reviewer 2 for his/her professional suggestion. Our previous investigations found that some NCIs are indeed involved as essential parameters for the catalytic cycle, especially for enantiocontrol situations. For the presented work, we highlight that the NCI between HBD moiety and the amide carbonyl is critical for the reaction process. And we indeed did not discuss other possible NCI-related issues. In the revised manuscript, we clarified this issue, and the reference mentioned by the reviewer was appended for the audience to understand the other possibilities further.

REVIEWERS' COMMENTS

Reviewer #1 (Remarks to the Author):

I reviewed the first version of the paper submitted by Chen, Huang and co-workers that reports "Amide C-N Bonds Activation by A New Variant of Bifunctional N-Heterocyclic Carbene". My requests were fully addressed and I am happy to support publication as is.